# Temperature evolution following joint loading promotes chondrogenesis by synergistic cues via calcium signaling

**Naser Nasrollahzadeh[1], Peyman Karami[1], Jian Wang[2], Lida Bagheri[1], Yanheng Guo[1], Philippe Abdel-Sayed[3], Lee Laurent-Applegate[3], Dominique P Pioletti[1]\***

[1]Laboratory of Biomechanical Orthopedics, Institute of Bioengineering, EPFL, Switzerland; [2]Institut des Matériaux et Institut des Sciences et Ingénierie Chimiques, Laboratoire des Polymères, Lausanne, Switzerland; [3]Regenerative Therapy Unit, Department of Musculoskeletal Medicine, Lausanne University Hospital, Lausanne, Switzerland

**\*For correspondence:** dominique.pioletti@epfl.ch

**Competing interest:** The authors declare that no competing interests exist.

**Abstract** During loading of viscoelastic tissues, part of the mechanical energy is transformed into heat that can locally increase the tissue temperature, a phenomenon known as self-heating. In the framework of mechanobiology, it has been accepted that cells react and adapt to mechanical stimuli. However, the cellular effect of temperature increase as a by-product of loading has been widely neglected. In this work, we focused on cartilage self-heating to present a 'thermo-mechanobiological' paradigm, and demonstrate how the coupling of a biomimetic temperature evolution and mechanical loading could influence cell behavior. We thereby developed a customized in vitro system allowing to recapitulate pertinent in vivo physical cues and determined the cells chondrogenic response to thermal and/or mechanical stimuli. Cellular mechanisms of action and potential signaling pathways of thermo-mechanotransduction process were also investigated. We found that co-existence of thermo-mechanical cues had a superior effect on chondrogenic gene expression compared to either signal alone. Specifically, the expression of Sox9 was significantly upregulated by application of the physiological thermo-mechanical stimulus. Multimodal transient receptor potential vanilloid 4 (TRPV4) channels were identified as key mediators of thermo-mechanotransduction process, which becomes ineffective without external calcium sources. We also observed that the isolated temperature evolution, as a by-product of loading, is a contributing factor to the cell response and this could be considered as important as the conventional mechanical loading. Providing an optimal thermo-mechanical environment by synergy of heat and loading portrays new opportunity for development of novel treatments for cartilage regeneration and can furthermore signal key elements for emerging cell-based therapies.

## Editor's evaluation

This is the first demonstration that temperature evolution in cartilage following joint loading alters chondrogenesis through activation of TRPV4 channels and downstream transcriptional effects. This new biology is consistent with the recently established role of temperature in theregulation of bone cells.

## Introduction

Mechanoregulation is a central process to maintain health and functionality of musculoskeletal tissues. Cells within the extracellular matrix (ECM) of load-bearing tissues receive various physical stimuli and maintain tissue homeostasis and function through reciprocal responses (*Ingber, 2003*; *Guilak et al., 2014*; *Humphrey et al., 2014*). The externally applied load generates different sensible cues within the cells surrounding microenvironments, such as spatiotemporal deformation, interstitial fluid pressure, shear stress, and ion mobility (*Guilak et al., 2014*; *Mow et al., 1999*). In this process, mechanical properties, organizational structure, and constituents of the ECM are massively influential and mediate the conveyed signals. Cell sensory systems integrate and convert the perceived physical inputs into the biochemical transients via different mechanisms of actions (e.g. calcium signaling; *O'Conor et al., 2014*). Accordingly, cells react and the elicited response modifies expression of genes and subsequent proteins to adapt tissue composition and structure (*Ingber, 2003*; *Jaalouk and Lammerding, 2009*). In particular, chondrogenic cells were shown to respond to physical stimuli when various dynamic loading regimens were applied (compression, torsion, tension, hydrostatic pressure) (*O'Conor et al., 2013*; *Salinas et al., 2018*) and when scaffold materials with different properties were used (elastic, viscoelastic, porous, non-porous) (*Panadero et al., 2016*; *Goetzke et al., 2018*). However, the role of multimodal physical cues and associated coupling phenomena in chondrogenesis still remains to be fully understood.

Articular cartilage is a load-bearing tissue with significant dissipative capacity. This property arises from the cartilage ECM intrinsic viscoelasticity and fluid-solid interactions (*Mak, 1986*; *Mow et al., 1992*). Dissipation is a function of the applied loading as well as material characteristics and is therefore an integrative variable for macroscopic physical cues generating spatiotemporal signals around cells. We have capitalized on this concept and demonstrated that matching dissipative capacity of scaffolds with native cartilage is an effective strategy to convey chondro-inductive cues to cells seeded in scaffolds under cyclic loading (*Abdel-Sayed et al., 2014a*; *Nasrollahzadeh et al., 2019*). Apart from direct effects of mechanical dissipation, being associated with matrix deformation, shear stress, and interstitial fluid pressure around cells, the cartilage dissipation may indirectly influence chondrocyte behavior by increasing tissue temperature. This phenomenon is called self-heating and occurs following conversion of the lost mechanical energy to heat over time (*Abdel-Sayed et al., 2014b*). In addition to cartilage dissipation, other factors such as ambient temperature and heat transfer from surrounding tissues could contribute to the temperature evolution in the joint. Previous studies indicated that cyclic compression exerted on cartilage could significantly increase its local temperature (*Abdel-Sayed et al., 2014b*; *Becher et al., 2008*; *Abdel-Sayed et al., 2013*). Notably, in vivo measurements inside intra-articular regions of knee joints have shown a temperature rise from 32°C (at rest) to ~38°C following 1 hr of jogging activity (*Becher et al., 2008*) at the ambient temperature of 19°C. Chondrocytes are sensitive to temperature and the literature suggests that their biophysical response to heat stimulation is dose-dependent (*Abdel-Sayed et al., 2013*; *Tonomura et al., 2008*; *Ito et al., 2015*). However, none of the conducted studies are based on a biomimetic temperature evolution regime. The loading-induced self-heating of cartilage, as shown in *Figure 1*, is an unexplored coupled phenomenon in the field of biomechanics which could initiate a new 'thermo-mechanobiology' paradigm.

In cartilage microenvironment, different elements residing inside and/or on plasma membrane of chondrocytes (cytoskeleton, nucleus, ion channels, primary cilium, integrin, and TGFβ₃ receptors) could be activated by receiving spatiotemporal cues and trigger intracellular changes (*O'Conor et al., 2013*; *Panadero et al., 2016*; *Loeser, 2014*). In particular, transient variation of intracellular calcium is a fundamental pathway for transduction of an external physical stimulus to control cellular functions (*Zhou et al., 2019*). It has been shown that, fluid flow (*Degala et al., 2011*), mechanical strain (*Pingguan-Murphy et al., 2006*; *Lv et al., 2018*), temperature (*Han et al., 2012*), and osmolarity (*O'Conor et al., 2014*) modulate cytosolic calcium oscillation in chondrocytes. Among different calcium mediators, transient receptor potential vanilloid 4 (TRPV4) is a multimodal ion channel that significantly contributes to the transduction of physical cues (*White et al., 2016*; *Lee et al., 2019*; *Servin-Vences et al., 2017*). The literature suggests that over-activation and full inhibition of TRPV4 channels in chondrocytes cause negative effects on joint development and cartilage homeostasis (*Krupkova et al., 2017*; *O'Conor et al., 2016*; *Atobe et al., 2019*; *Clark et al., 2010*). Of note, TRPV4 channels are responsive to mechanical loading (*O'Conor et al., 2014*; *Lv et al., 2018*) and

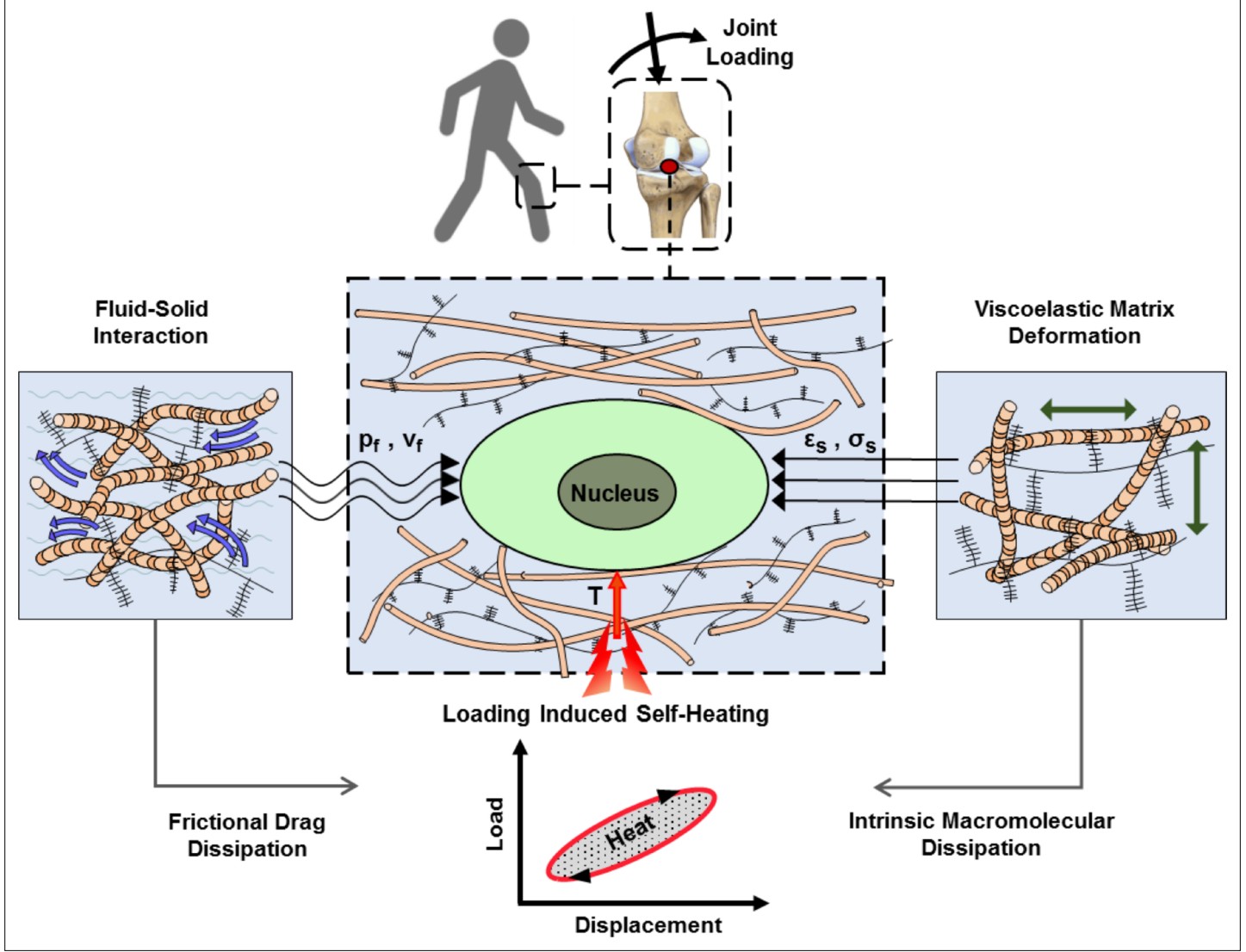

**Figure 1.** Cartilage loading-induced self-heating during physical activity. Mechanical hysteresis in cartilage tissue during joint loading generates heat and causes the temperature rise over time. The self-heating of cartilage originates from both, intrinsic matrix viscoelasticity and solid-fluid interaction sources during joint loading. Structural and material characteristics of matrix microenvironment are acting in concert with external stresses to provide direct and indirect biophysical cues for cells such as deformation and temperature variation ($p_f$, $v_f$: fluid pressure and flow velocity; $\varepsilon_s$, $\sigma_s$ solid matrix strain and stress; T: temperature).

temperature rise (*Güler et al., 2002*; *Gao et al., 2003*), making it a potential sensory system for translation of stimuli involved in loading-induced self-heating phenomenon.

The overall objective of the present work is to address cartilage thermo-mechanobiology. For this purpose, we first developed an original model system capable of simulating the self-heating phenomenon in vitro. Specifically, our customized system encompassed a modular bioreactor designed to independently control the evolution of temperature and applied mechanical loading on cell-laden constructs, as well as a biomechanically functional scaffold recapitulating cartilage viscoelastic properties. Having developed the tailored in vitro platform, we then determined the individual contribution and the combined effects of temperature and loading on cell response. In this study, the cellular reaction was evaluated through varied expression of chondrogenic genes following an applied stimulus. Finally, we aimed to identify downstream signaling and the potential role of TRPV4 ion channels as a mediator in the thermo-mechanotransduction process. Accordingly, the influence of thermo-mechanical cues on cell response was investigated when inhibiting TRPV4 channels as a signal transducer or diminishing cytosolic calcium variation as a mechanism of action. An insight on the effect of

cartilage microenvironment on chondrocyte behavior and respective transduction cascade can positively contribute to the development of emerging cell-based therapies for tissue regeneration.

## Results

### A custom-made in vitro platform for cartilage thermo-mechanobiology

We developed an in vitro platform allowing to recapitulate loading-induced self-heating in articular cartilage and study corresponding cell response. Our system consisted of two essential components: (i) a poro-viscoelastic scaffold, with comparable dissipative capacity and equilibrium compressive stiffness to cartilage, seeded with human chondro-progenitor cells (ECPs); (ii) a modular bioreactor to independently control applied mechanical loading, temperature increase, as well as gas concentration and humidity levels during stimulation. We have used ECPs because they provide an off-the-shelf cell source and have potential chondro-differentiation capability and reportedly they can produce cartilage matrix in vivo (*Studer et al., 2017*). By using this cell source, we were able to have a standardized platform for all of our experimentation with the high quality of cells and full safety features for experimental models (*Laurent et al., 2020*; *Laurent et al., 2021*).

As a 3D support for cells, a fatigue-resistant hydrogel (*Figure 2a, b*) was developed by combination of flow-dependent and flow-independent dissipation sources to recapitulate cartilage viscoelastic properties. This was accomplished by a hybridly crosslinked (combination of covalent and physical bonds) pHEMA network and a poorly permeable macrostructure (*Figure 2—figure supplement 1*). Rearrangement of polymeric chains, dissociation of reversible bonds, and fluid frictional drag mechanisms contributed to viscoelastic behavior of the developed hydrogel as described in detail elsewhere (*Nasrollahzadeh et al., 2019*). The stress-strain curve under cyclic loading was minimally varied showing that the developed hydrogel can preserve applied mechanical cues during multiple physiological stimulations. The poro-viscoelastic scaffold was also functionalized by RGD peptides (*Figure 2—figure supplement 2*) to enhance cell attachment and cell-matrix interactions during loading. The X-ray photoelectron spectroscopy (XPS) survey scan recorded on pure (non-modified) and RGD functionalized samples (*Figure 2c*), confirmed the effectiveness of the RGD grafting process. As the pure pHEMA hydrogel does not contain any nitrogen atom in its molecular structure, no signal was observed in respective range for pure samples contrary to RGD functionalized samples with a peak for nitrogen at 402 eV.

In addition, cell proliferation and DNA content (*Figure 2—figure supplement 3*) of the functionalized constructs were significantly higher than the pure samples without RGD motifs. We could minimize the destructive effect of the bio-conjugation process on the crosslinked network and could maintain the dissipative capacity and equilibrium stiffness of the functionalized hydrogels in the range of cartilage tissue (*Mak, 1986*; *Mow et al., 1992*; *Abdel-Sayed et al., 2014a*; *Abdel-Sayed et al., 2014b*) after modification (*Figure 2d*). While we could not observe a noticeable change on the surface morphology of the polymeric scaffolds after functionalization process, the SEM images showed that cells could spread and attach to multiple binding points on the pores of hydrogels with RGD motifs (*Figure 2—figure supplement 3*). We also observed more detached cells in non-modified hydrogels without endogenous binding sites for cells when cyclic compression was combined with temperature rise (*Figure 2—figure supplement 4*).

Apart from tailored biomaterial's properties, reliable control over applied stimuli are required to understand the effects of loading-induced self-heating on chondrogenesis. Specifically, we need to simulate the cells thermal and/or mechanical environment during joint loading. We custom designed our bioreactor system as shown in *Figure 2f*, employed an adjustable loading mechanism, and applied a convection dominant heat transfer system inside the wells to ensure consistent stimulation of samples. The performance of the developed bioreactor in different modes of action relevant to our study was then validated (*Figure 2—figure supplement 5*). Briefly, we confirmed that samples in different wells receive similar external stimuli according to the defined operation protocol. The controlled culture variables were verified by external calibrators. The temperature increase was modeled by *Equation 1* and its parameters were estimated based on reported in vivo data during jogging (*Becher et al., 2008*) using the least square method.

$$T(t) = A + B\left(1 - e^{-ct}\right) \tag{1}$$

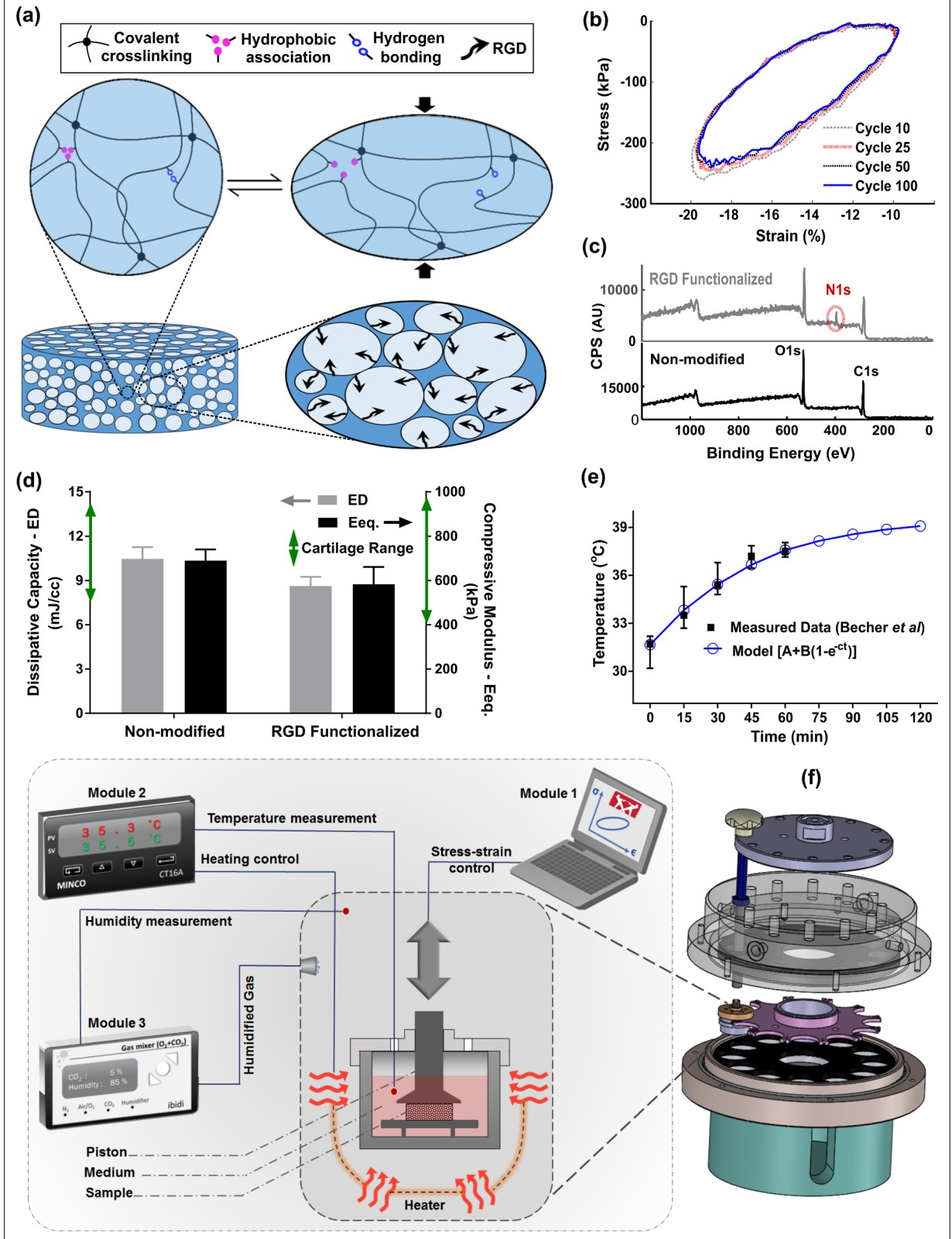

**Figure 2.** Developed in vitro system to study cartilage thermo-mechanobiology. (**a**) Poro-viscoelastic hydrogel with RGD decoration on pores. The polymeric network of the scaffold is hybridly crosslinked with physical hydrophobic associations, hydrogen bonds, and covalent crosslinks between chains. Following an applied deformation, flexible network reorganization and fluid-solid interaction within the porous structure occur resembling cartilage dissipative behavior. (**b**) The hysteresis loop following loading and unloading steps was preserved over different cycles indicating fatigue

*Figure 2 continued on next page*

*Figure 2 continued*

resistance capability of the hydrogel. The shrinkage of the loop is mainly at the onset and remains stable after preconditioning cycles, owing to reversible sources of dissipation. (**c**) X-ray photoelectron spectroscopy (XPS) survey scan of pHEMA porous hydrogels before and after RGD functionalization. The appearance of an N1s peak at 400 eV in the XPS spectra of functionalized samples is the evidence for a successful binding of RGD peptides to the exposed hydroxyl groups of porous pHEMA hydrogel. (**d**) Mechanical properties of the pure (non-modified) and RGD functionalized hydrogels and reported range of cartilage properties (green arrows) in literature. (**e**) The measured intra-articular temperature following jogging activity as reported in *Becher et al., 2008*, and exponential fitted curve to predict the temperature evolution. (**f**) Modular and custom-designed bioreactor. The apparatus consists of adjustable loading system with embedded screws, chamber cap with different inlets/outlets, culture wells and pistons, wells carrier, chamber base and support. The left schematic illustrates conceptual design of the modular bioreactor for thermo-mechanical stimulation of cell-laden hydrogels.

The online version of this article includes the following figure supplement(s) for figure 2:

**Figure supplement 1.** Morphological structure of the dissipative hydrogel.

**Figure supplement 2.** Applied two-step bio-conjugation process (hydroxyl group activation and RGD peptide grafting) for functionalization of pHEMA-based hydrogels.

**Figure supplement 3.** Cell attachment, proliferation, morphology, and distribution inside pure (non-modified) and RGD functionalized hydrogels.

**Figure supplement 4.** Cell viability and attachment before and after thermo-mechanical stimulation inside the bioreactor for non-modified (top) and RGD functionalized (bottom) hydrogels.

**Figure supplement 5.** Function evaluation of the developed bioreactor for simulation of cartilage loading-induced self-heating.

**Figure supplement 6.** Heat transfer simulation in one culture well of the bioreactor by constant temperature boundary condition (32°C) at the outer interface.

**Figure supplement 7.** Heat transfer simulation in one culture well of the bioreactor by application of adiabatic boundary condition.

The optimally fitted temperature evolution function (A = 31.62°C, B = 7.99°C, and c = 0.023 s$^{-1}$) predicted around 7°C temperature rise after 2 hr (*Figure 2e*) with a good correlation coefficient (R = 0.95). We controlled the evolution of the culture temperature inside the wells based on the model prediction by an external heat source to simulate intra-articular knee temperature during joint loading. Given the volume of the culture medium in each well, the size of the sample and the system boundary conditions, the accumulated lost energy from the dissipative hydrogel negligibly changed the culture medium temperature (<0.5°C). Our heat transfer simulation confirmed this observation when the well interface temperature was kept constant at 32°C (*Figure 2—figure supplement 6*). However, in an unrealistic adiabatic condition, our numerical simulation showed a temperature rise (*Figure 2—figure supplement 7*) due to hydrogel's dissipative capacity. The isolated temperature and loading control strategy in our bioreactor design allowed us to decompose direct and indirect effects of mechanical loading on cell responses during self-heating of cartilage.

## Temperature evolution following mechanical loading optimally promotes chondrogenesis by simulating the cartilage native environment during activity

We first evaluated the viability of human chondro-progenitor cells (origin and characterization described elsewhere; *Darwiche et al., 2012*) within the functionalized hydrogels at two different culture temperatures. The cell-seeded hydrogels were cultured for 12 days at knee intra-articular temperature at rest (32.5°C) and the conventional in vitro condition (37°C) corresponding to the core body temperature. We confirmed stable attachment, high viability, and normal proliferation of distributed cells within the RGD functionalized hydrogels (*Figure 3a-c*). Our results indicated that cells metabolic activity, the amount of extracted DNA and RNA (*Figure 3—figure supplement 1a*) were slightly higher at 32.5°C compared to 37°C. These data suggest that adjusting the culture temperature at 32–33°C suits human chondro-progenitor cells metabolism which is in agreement with the reported results for human and porcine chondrocytes in monolayer and pellet cultures (*Ito et al., 2015*; *Ito et al., 2014*).

To understand the influence of temperature and mechanical load on biological response, gene expression data was compared by applying coupled or isolated signals (*Figure 3—figure supplement 2*). Our findings demonstrated that the loading-induced self-heating (simultaneous temperature evolution and mechanical loading) positively enhances chondrogenic expression of cells and outperforms mechanical or thermal stimulus alone (*Figure 3d-i*). Importantly, we obtained a boosted effect on the expression of Sox9, as shown in *Figure 3d*, following biomimetic thermo-mechanical stimulation

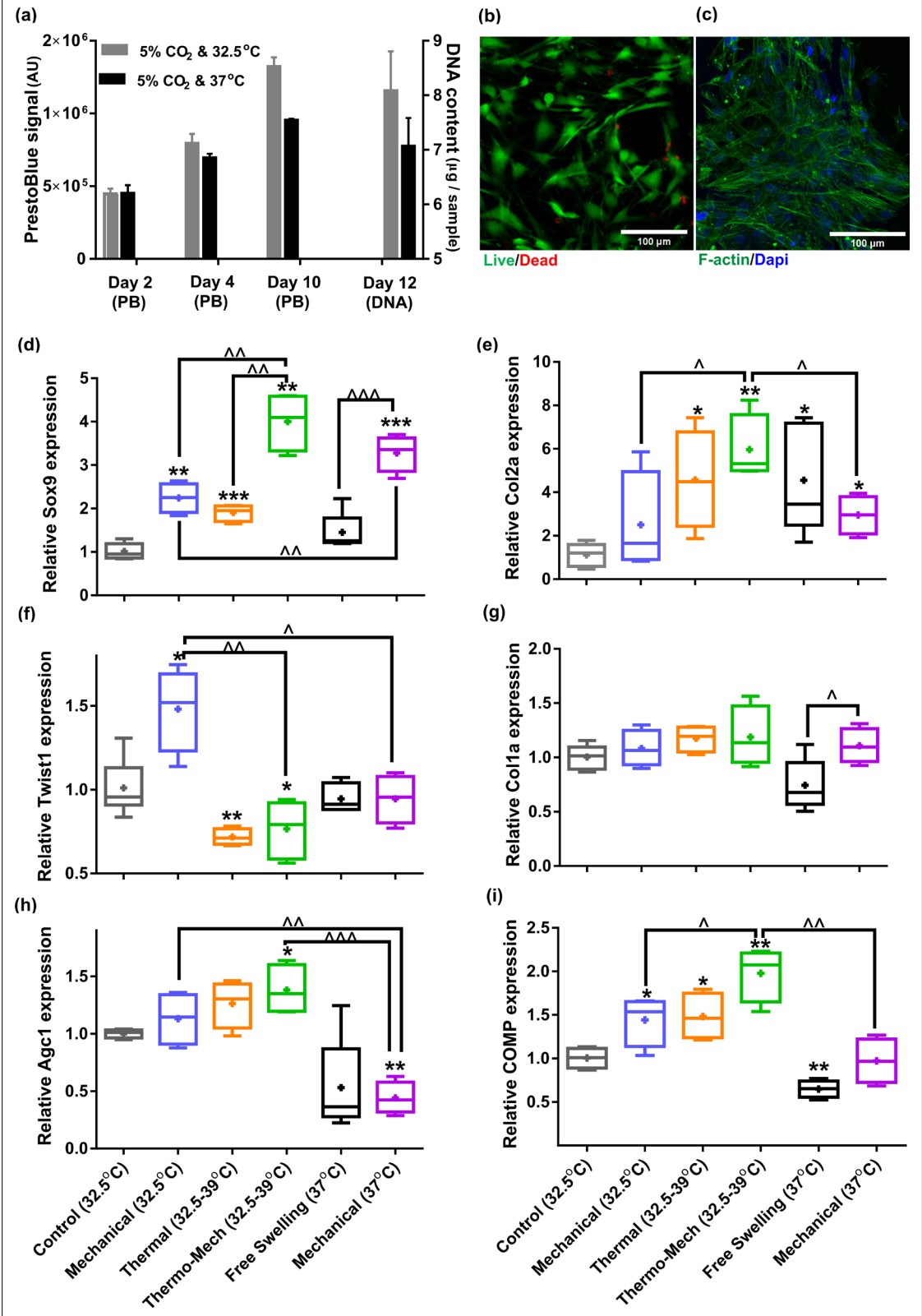

**Figure 3.** Human chondro-progenitor cells behavior inside functionalized dissipative porous hydrogels. (**a**) Metabolic activity (obtained by PrestoBlue [PB] assay) and DNA content of free swelling cell-laden hydrogels in different culture temperatures corresponding to knee intra-articular temperature at rest and core body temperature. (**b**) Live/dead labeling of cells inside porous hydrogels 4 days post seeding cultured at 32.5°C (green stain demonstrates live cells and red stain shows dead cells). (**c**) Actin filament and nucleus immunostaining of cells, 6 days after seeding. (**d:i**) Comparison of

*Figure 3 continued on next page*

*Figure 3 continued*

the relative gene expression of cells in response to different biophysical cues applied in 3 alternate days for 90 min normalized to control free-swelling samples cultured at 32.5°C (significant differences with control group (*) or between specific groups (^) p < 0.05; n = 4–6). Overall the results indicate that loading-induced self-heating, designated as Thermo-Mech (32.5–39°C), optimally promoted chondrogenesis.

The online version of this article includes the following source data and figure supplement(s) for figure 3:

**Source data 1.** Fold change of samples in different studied groups.

**Figure supplement 1.** Extracted RNA and Col2/Col1 fold-change ratio.

**Figure supplement 2.** Schematic workflow of the performed in vitro thermo-mechanobiological experiment.

designated as Thermo-Mech (32.5–39°C). Sox9 is a transcription factor and its role as one of the key regulators of chondrogenic differentiation processes is well established in the literature (*Akiyama et al., 2002*). Upregulation of Sox9 gene under thermo-mechanical stimulation was significantly higher than the control, mechanical, and thermal groups. In parallel, Twist-related protein 1 (Twist1), which is known as an inhibitor of chondrogenesis (*Gu et al., 2012*), exhibited a significantly downregulated expression following the thermo-mechanical stimulation (*Figure 3f*). Highest expression was also observed for functional chondrogenic markers, Col2a1, Agc1, and COMP, when cells received the combined physical cues. Col1a1 expression (*Figure 3g*) was not significantly different between stimulated and control groups. The obtained relative fold-change ratio for Col2a1/Col1a1 (*Figure 3—figure supplement 1b*) confirms enhanced chondrogenic differentiation capacity regardless of the absolute level of Col2 and Col1 genes. Collectively, the applied thermo-mechanical stimulus provided a chondro-inductive signal by simulating the cartilage native environment under loading.

While still a promoter of chondrogenesis, isolated mechanical or thermal stimuli were not as effective as thermo-mechanical stimulus combined. Notably, application of intermittent temperature increases (from 32.5°C to 39°C during 1.5 hr) on cells significantly upregulated expression of Sox9, Col2a, COMP and downregulated Twist1. Unlike traditional mechanobiological in vitro models with unrealistic culture temperature of 37°C, we evaluated the effect of steady-state incubation temperature on chondrogenic cellular responses. Compared to the control group, a significant fold increase was detected on expression of COMP by loading at 32.5°C. An enhanced expression of Col2a was also observed following mechanical stimulation without temperature variation. A statistically significant difference in fold increase of Sox9 was obtained between stimulated samples at 37°C and 32.5°C. This implies that chondrogenic cells respond better to compressive loading at higher temperatures which supports the common mode of action in most bioreactors for cartilage tissue engineering. Yet, simulation of loading-induced self-heating condition better promoted chondrogenic markers compared to mechanical stimulus at constant temperatures (32.5°C or 37°C). This indicates that cells could sense the transient temperature rise. The transient effect should therefore be considered as a potential regulatory cue besides external stress.

## Cartilage thermo-mechanotransduction is a TRPV4-mediated and calcium-dependent signaling mechanism

Knowing that oscillation of intracellular calcium is an important regulator of chondrogenesis, the role of $Ca^{2+}$ signaling in thermo-mechanotransduction process was evaluated. We also examined the contribution of multimodal TRPV4 ion channels to this process as a potent $Ca^{2+}$ modulator (*Figure 4a*). We have measured time-dependent gene expression of TRPV4 channels in the scaffolds and confirmed their functional expression on human chondro-progenitor cells over our study period (*Figure 4—figure supplement 1*). When cells cultured in scaffolds without any stimulation at 32.5°C, we observed downregulation of the TRPV4 genes at days 6 and 10 compared to day 2. However, a significant upregulation in expression of TRPV4 gene was detected by application of biomimetic temperature evolution, mechanical loading, and combination thereof. Immunostaining of cells using a specific antibody confirmed the presence of TRPV4 channels on cells (*Figure 4b*) seeded in functionalized hydrogels. The functioning of TRPV4 was confirmed by live florescence imaging of cells following administration of 10 nM TRPV4 agonist, GSK1016790 (GSK101). We could detect multiple dynamic peaks after GSK101 delivery (following 50-70 s baseline recording) when analyzing intracellular calcium signals as shown in *Figure 4c* for three representative cells (see *Video 1* for the complete response of a group of cells). We then verified significant inhibition of TRPV4-mediated calcium

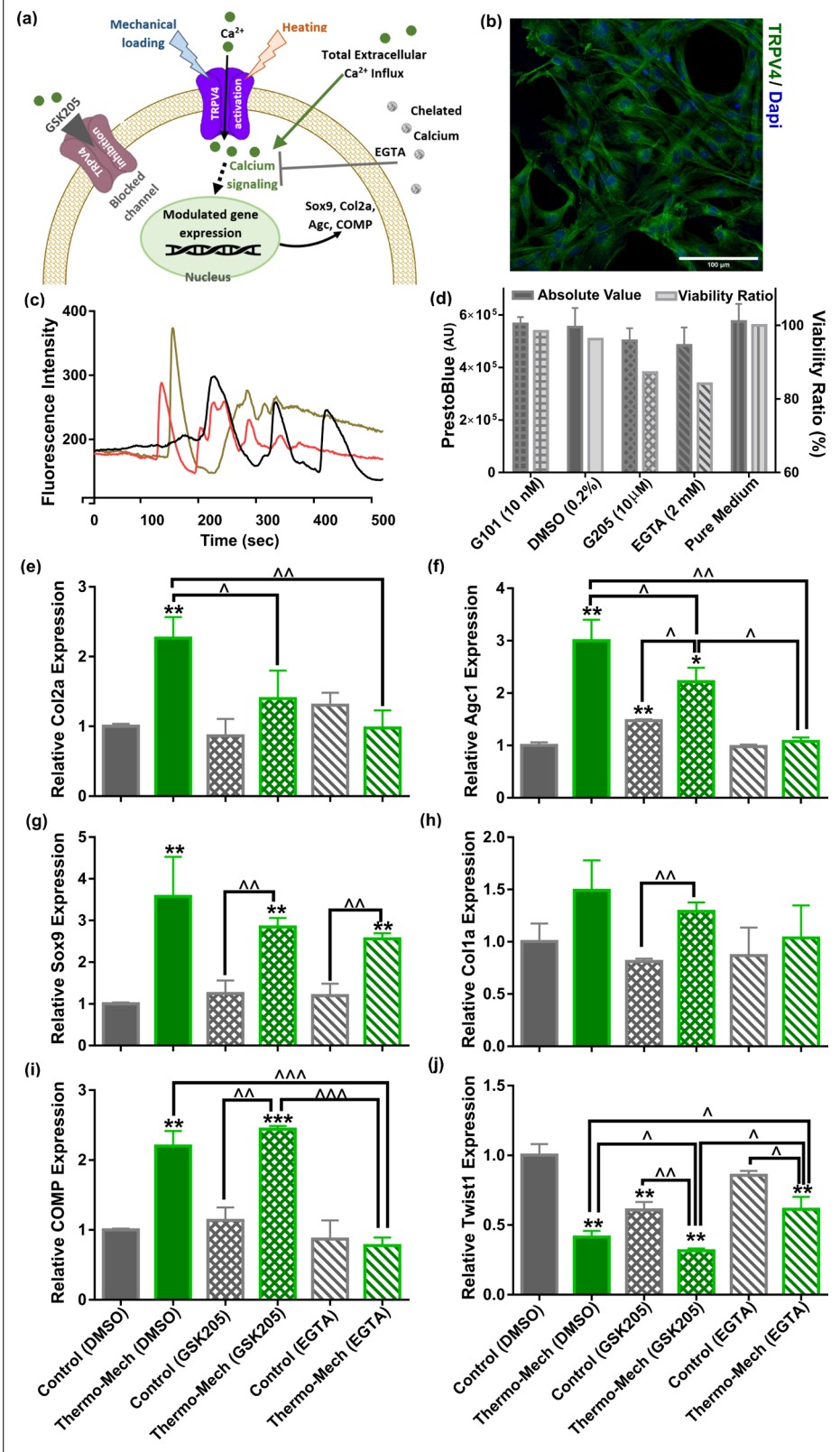

**Figure 4.** Calcium signaling mechanism, functional characterization of transient receptor potential vanilloid 4 (TRPV4) ion channels expression in human chondro-progenitor cells, and gene expression results with manipulated pathways regulating cytosolic $Ca^{2+}$ variation. (**a**) Schematic of TRPV4-mediated calcium signaling for transduction of thermo-mechanical cues which could be inhibited by GSK205 (TRPV4 antagonist). Generally, the intracellular

*Figure 4 continued on next page*

*Figure 4 continued*

calcium can be varied through different pathways and all of them are affected when extracellular Ca²⁺ ions are chelated with EGTA. (**b**) Successful binding of the specific-TRPV4 antibody to human chondro-progenitor cells confirmed expression of TRPV4 channels on cells seeded in porous hydrogels. (**c**) Response of cells to TRPV4 agonist (10 nm GSK101) confirmed functionality of the gates for modulation of intracellular calcium content. (**d**) Viability of cells in presence of pertinent agonsit/antagonist of calcium pathways. (**e:j**) Relative gene expression of cells in different groups normalized to free-swelling control group with DMSO as drug carrier when employing specific antagonist for inhibiting TRPV4 channels (GSK205) or chelating extracellular calcium sources (EGTA). The control groups are free-swelling samples cultured at 32.5°C and stimulated samples receiving combined thermo-mechanical cues at the same culture medium containing DMSO, GSK205, or EGTA (* significant differences with control group with p < 0.05; ^ significant differences between specific groups with p < 0.05; n = 3–4). Extracted results indicated that thermo-mechanotransduction signaling cascade was almost incomplete without extracellular calcium sources and TRPV4 channels played a key role in translation of perceived physical cues to calcium transients.

The online version of this article includes the following source data and figure supplement(s) for figure 4:

**Source data 1.** Fold change of samples in different groups of thermo-mechanotransduction pathways study.

**Figure supplement 1.** Transient receptor potential vanilloid 4 (TRPV4) expression of chondro-progenitor cells is time and stimuli dependent.

**Figure supplement 2.** Gene expression results indicated that transient receptor potential vanilloid 4 (TRPV4) acts as a signal integrator for thermal and mechanical stimuli.

signaling following 10 nM GSK101 delivery (*Video 2*) by using 10 µM TRPV4 antagonist (GSK205) in culture medium 1 hr before imaging. Moreover, full chelation of extracellular calcium sources was obtained by employing 2 mM EGTA in the culture medium which completely diminished cytosolic calcium variation. We also evaluated the cell viability in presence of effective quantities of employed chemicals (DMSO as carrier, EGTA, GSK101 and GSK205 as agonist/antagonist) in our pathway study. Results of the assay (*Figure 4d*) for all tested conditions showed more than 85% viability at day 8 after using chemicals in culture medium for 4 hr during 2 different days.

The inhibition of TRPV4 channels with 10 µm GSK205 diminished the elevated expression of the main chondrogenic markers when cells received the combined physical cues. As shown in *Figure 4e-h*, a significant downregulation for expression of Col2a and Agc1 genes was obtained in stimulated samples with TRPV4 antagonist. We also observed a slight decrease in Sox9 and Col1a expression

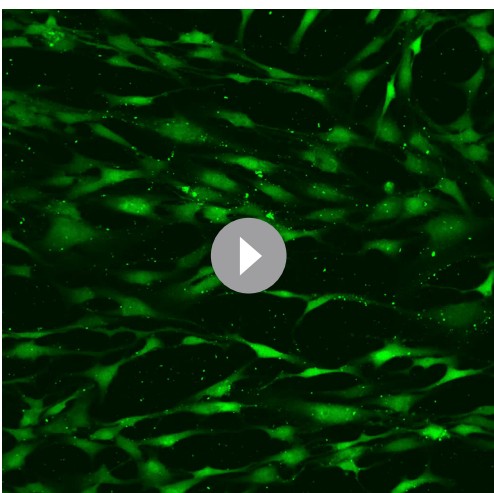

**Video 1.** Transient receptor potential vanilloid 4 (TRPV4) channel activation in human chondro-progenitor cells cultured in 2D by using 10 nM GSK01 agonist. Most of the cells are responsive to delivery of TRPV4 activator.

https://elifesciences.org/articles/72068/figures#video1

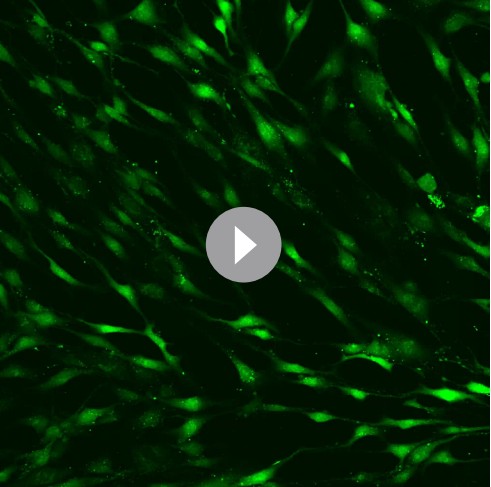

**Video 2.** Transient receptor potential vanilloid 4 (TRPV4) channel inhibition in human chondro-progenitor cells in 2D by using 10 µM GSK205 antagonist. Only a few cells are responsive to delivery of TRPV4 activator.

https://elifesciences.org/articles/72068/figures#video2

with blocked TRPV4 channels. Inhibiting TRPV4 activity in free-swelling samples did not affect relative expression of the Col2a, Sox9, Col1a, and COMP (*Figure 4i*). In contrast, we could detect a significant downregulation of Twist1 in control samples with GSK205 (*Figure 4j*). Consequently, further downregulation was obtained for Twist1 gene by thermo-mechanical stimulation when TRPV4 activity was inhibited. Our results also indicated that TRPV4 acts as signal integrator for thermal and mechanical stimuli (*Figure 4—figure supplement 2*). By comparing the Col2a and Sox9 gene expression for either cue alone, with and without GSK205, one can observe reduced levels of these markers when TRPV4 is inhibited. The downregulation in blocked TRPV4 channels condition is statistically significant for Sox9 only in case of mechanical loading and for Col2a only in case of thermal stimulus.

Without application of external physical stimulus, we could not observe a significant change on relative gene expression of cells in presence of 2 mM EGTA. Similarly, the promoted expression of chondrogenic markers following thermo-mechanical stimulation was almost abrogated without external $Ca^{2+}$ sources. In particular, we measured a significant reduction in expression of Col2a, COMP, and Agc1 genes and a significant increase in expression of Twist 1 (*Figure 4e, f, i, j*). The average Sox9 expression (*Figure 4g*) was also decreased; however, the downregulation was not statistically significant in the absence of external $Ca^{2+}$. Taken together, our results suggest that TRPV4 mediates thermo-mechanotransduction processes and the Ca2+ signaling participates more in modulation of functional ECM markers (collagen type 2, cartilage oligomeric matrix protein, and aggrecan genes) than Sox9 transcription factor.

## Discussion

Many groups, including ours, have already demonstrated the positive impact of mechanical loading on chondrogenesis in explant and cell-scaffold constructs (*O'Conor et al., 2014*; *O'Conor et al., 2013*; *Salinas et al., 2018*; *Nasrollahzadeh et al., 2019*; *Schätti et al., 2011*; *Gan et al., 2018*; *Anderson and Johnstone, 2017*). However, previous in vitro mechanobiological studies typically considered a constant culture temperature of 37°C which does not correspond to the physiological in vivo condition in human knee. For the first time, we investigated cartilage thermo-mechanobiology and demonstrated the synergetic effect of heat and loading stimuli at the transcriptional level of chondrogenesis. We further demonstrated that the thermo-mechanotransduction is indeed calcium dependent and TRPV4 gates on the plasma membrane play a key role in this process. This study also revealed that a decomposed thermal cue, as a by-product of loading, could itself promote chondrogenesis similar to or even better than the mechanical signal alone. Our findings support the concept that ECM viscoelasticity can indirectly influence biological response of cells as self-heating phenomenon links the temperature evolution to cyclic loading of viscoelastic materials.

### Simultaneous consideration of inter-related cues for cartilage thermo-mechanobiology

As already well established, the biophysical cues significantly influence the cellular responses in musculoskeletal tissues (*Ingber, 2003*; *Guilak et al., 2014*; *Humphrey et al., 2014*; *Jaalouk and Lammerding, 2009*). In a mechanistic view point, structural and material properties of ECM microenvironment are acting in association with external stresses to generate physical cues for cells (*Humphrey et al., 2014*; *Panadero et al., 2016*; *Goetzke et al., 2018*). The cartilage dissipative capacity originates from intrinsic matrix viscoelasticity and solid-fluid interactions under an applied loading. These dissipative sources are associated with the matrix deformation, hydrostatic pressure, and interstitial fluid flow around cells which were previously considered as mechanobiological regulators (*Mow et al., 1999*; *O'Conor et al., 2013*; *Salinas et al., 2018*). Cartilage dissipation, therefore, could be a functional property for chondrogenesis overarching all solid- and fluid-related mechanical signals (*Abdel-Sayed et al., 2014a*; *Nasrollahzadeh et al., 2019*). In parallel, energy dissipation in avascular cartilage during joint loading generates heat and causes the temperature evolution over time (*Abdel-Sayed et al., 2014b*). The absolute temperature rise in cartilage microenvironment also depends on the joint boundary condition (e.g. the ambient temperature and heat flux by surroundings tissues). Regardless of the heat source(s) and of the complex heat transfer mechanisms inside articular cavity, the varied temperature during joint loading could affect chondrocyte biological responses. To simulate such conditions and study cartilage thermo-mechanobiology, we developed a fatigue-resistant

dissipative hydrogel for 3D culture of cells and a modular bioreactor to apply respective biophysical cues in vitro as there was not an available platform for this purpose elsewhere.

The developed modular bioreactor is necessary for evaluating the role of the temperature rise during mechanical loading in a laboratory design. Yet, the use of human articular cartilage explants instead of cell-seeded constructs in our customized bioreactor could better reproduce the in vivo thermo-mechanical environment. However, the difficulty in obtaining healthy human articular cartilage with similar size and physiological state was the main reason for which we developed a reproducible model and used cell-laden hydrogels. Indeed, this system has its own limitation and the spread morphology of the cells in our dissipative porous hydrogels is one of them. The ideal construct would be a mechanically capable scaffold that provides an environment for moderate cells adhesion and fairly maintains round morphology of cells. This might be realized by cells encapsulation into an adhesive soft hydrogel (*Karami et al., 2021*) with endogenous binding sites for cells which is firmly integrated in a stiff and porous 3D support with load-bearing capacity (*Liao et al., 2013*; *Boere et al., 2014*). The limitation of our porous dissipative hydrogel for this method of construct preparation is its intentional low permeability (order of $E^{-14}$ m$^2$) to induce frictional drag dissipation. The permeability can be increased by enlarging the size of the pores and providing full interconnectivity within the porous structure. Nonetheless, recapitulating cartilage dissipation might not be achievable in this condition. The focus of current research was studying the effect of loading-induced temperature evolution on cells behavior by a reasonable and reproducible in vitro model. Given the fact that we employed identically prepared cell-scaffolds constructs for all of our experimentation, we assumed that the isolated role of externally applied stimuli on cells chondrogenic response could be evaluated irrespective of cells morphology.

The reciprocal relationship between cells and ECM governs cartilage hemostasis as chondrocytes receive physical cues and regulate tissue turnover to maintain its functional characteristics (e.g. dissipation). An altered gene expression is generally considered as the first indicator for cells responsiveness to conveyed cues which could ultimately lead to long-term matrix deposition depending on other contributing factors. Collagen type II (Col2a), aggrecan (Agc1), and SRY-related HMG-box gene 9 (Sox9) genes are the most utilized chondrogenic markers at the molecular level for mechanobiological studies (*O'Conor et al., 2013*; *Anderson and Johnstone, 2017*). Not only does Sox9 contribute to early chondrogenesis and matrix synthesis (*Akiyama et al., 2002*), but it also inhibits chondrocyte terminal differentiation (*Leung et al., 2011*). Col2a and Agc1 are anabolic markers of main ECM components in cartilage tissue and COMP gene controls synthesis of the linkage protein with multiple binding possibilities to integrate and stabilize the ECM (*Koelling, 2006*). Twist1 is known to hinder chondrogenesis by direct inhibition of Sox9 as the master regulator of the process (*Gu et al., 2012*). Our customized model system enabled us to show that chondrogenic cells immediately responded to biomimetic thermo-mechanical cues by altering their gene expression and that cells are able to sense the transient temperature rise. We evaluated cellular reactions after three rounds of stimulation to consider their probable desensitization and estimate a robust transcriptional response. However, given the transient nature of the chondrogenic markers to an applied stimulation (*Scholtes et al., 2018*), analysis of results at different time points could help to better understand the gene expression profile. Additionally, it still remains to be determined to what extent the promoted chondrogenic gene expression could enhance downstream protein expression and functional properties.

## Cartilage thermo-mechanics portrays new opportunity to advance chondrogenesis in vitro

The expression of chondrogenic markers was enhanced by the applied physical stimulus in all groups except for Agc1 and COMP in one group (*Figure 3*). The obtained gene expression data for Col2a1, Agc1, Col1a, Sox9, Twist1, and COMP clearly show that thermo-mechanical stimulus generates more potent signaling to elicit cell responses compared to either cue alone. Moreover, Col2a1/Col1a relative fold-change ratio confirmed that combined biomimetic stimuli enhanced progenitor cell chondrogenic capacity. Yet, prolonged culture of ECP cells with growth factor supplements (at least 21 days) was needed to reach mature chondrogenic differentiation state where the absolute expression of Col2 could exceed the absolute expression of Col1 as reported by other studies with this cell line (*Studer et al., 2017*; *Laurent et al., 2021*). One compelling aspect of this finding is that employing bioreactors which are capable of controlling temperature evolution during dynamic loading should be preferred

for tissue engineering applications. Indeed, chondro-inductivity can be enhanced by the condition which simulates the in vivo cartilage more similarly as shown by other multi-functional bioreactors (*O'Conor et al., 2013*; *Schätti et al., 2011*). For instance, simulating complex physiological loading conditions in articulating joints by combined compression, sliding, and shear was shown to be more effective than a single loading mode (*Schätti et al., 2011*). In addition, combining hypoxia culture with multi-directional loading performed better than normaxia (*Wernike et al., 2008*) as low oxygen level could direct chondrogenesis. Accordingly, investigating controlled oxygen tension combined with thermo-mechanical stimulation might shed more light on the in vivo chondrogenic process.

Dynamic temperature evolution also had promising impact on transcription of chondrogenic markers in the current study where applied thermal stimulus outperforms pure mechanical loading. Significant upregulation in Col2a1, COMP, and Sox9 expression as well as downregulation of Twist1 gene clearly indicate enhanced cellular responses to isolated self-heating cue (temperature rise without loading). In line with our findings, available evidence in the literature suggests that proper heat stimulation can enhance and accelerate chondrogenic differentiation processes (*Tonomura et al., 2008*). Over stimulation by high temperature, however, reduces the cell viability and inhibits cartilage matrix synthesis (*Ito et al., 2015*). Therefore, it is important to simulate the dynamics of thermal environment in vivo and optimize the influence of thermal stimuli by defining a biomimetic intermittent temperature rise. Considering clinical translation, it is encouraging to have chondro-inductive cues only with application of careful heating to simulate joint thermal environment during physical activity. Such stimulus can be simply delivered to patients in need by development of a customized heat therapy device for joint disease. It remains to elucidate if this argument is true for osteoarthritic cartilage as damaged chondrocytes may respond differently to biophysical cues as reported recently (*Agarwal et al., 2021*).

## Calcium signaling is a major mechanism of action in transduction of thermo-mechanical cues sensed by TRPV4 channels

Understanding the translation mechanisms of external stimuli and their interaction with molecular signaling pathways is essential for development of therapeutic products to modulate the target regulators. It is well established that TRPV4 is a key mediator in transduction of mechanical signals in chondrocytes which is necessary for maintenance of cartilage tissue (*O'Conor et al., 2014*; *White et al., 2016*; *Agarwal et al., 2021*). In parallel, the reported activation temperature around 30°C for TRPV4 (*White et al., 2016*; *Güler et al., 2002*) makes this channel a potential mediator of loading-induced self-heating in cartilage. Our study revealed that TRPV4 is a potent modulator for expression of Col2a and Agc1 genes in response to thermo-mechanical cues. These results are in agreement with the data reported for post-transcriptional markers determined by protein expression and functional properties of cell-encapsulated hydrogels under mechanical stimulation (*O'Conor et al., 2014*; *Gan et al., 2018*). The obtained gene expression profile of Sox9 in this study suggested a partial contribution of TRPV4 to its transcription in response to thermo-mechanical cues. This means that there are parallel mechanisms for sensing the applied stimuli and eliciting Sox9 enhancement which could be other transmembrane channels, primary cilium or integrin (*O'Conor et al., 2013*; *Panadero et al., 2016*; *Lv et al., 2018*; *Krupkova et al., 2017*). Integrin-mediated pathways arising from cell-matrix interaction are mechanisms by which cells may sense the mechanical cues in the surrounding microenvironment (*Humphrey et al., 2014*; *Loeser, 2014*). Since RGD motifs were conjugated onto the pores of the developed hydrogels, a stable and effective cell-matrix interaction during loading can be expected. According to our results, significant upregulation of Sox9 in the mechanically loaded samples with and without GSK205 (*Figure 4—figure supplement 2*) could imply the contribution of an integrin pathway. Of note, previous studies also showed that integrin could regulate activation of ion channels in chondrocytes and cell-matrix interactions could modulate calcium signaling dynamics in response to physical cues (*Degala et al., 2011*; *Han et al., 2012*; *Jablonski et al., 2014*). We could not find any direct thermo and/or mechano-biological study in the literature reporting even partial effect of TRPV4 blocking on Sox9 expression in chondrogenic cells. However, in agreement with our findings, in vivo and in vitro studies have shown that pharmacological activation/inhibition of TRPV4 channels could mediate Sox9 expression (*Atobe et al., 2019*; *Muramatsu et al., 2007*).

TRPV4 activity can modulate intracellular calcium transients and literature findings indicate that a well-adjusted TRPV4-mediated $Ca^{2+}$ signaling is necessary for correct biological functioning of chondrocytes (*O'Conor et al., 2014*; *Krupkova et al., 2017*; *O'Conor et al., 2016*; *Atobe et al., 2019*;

*Clark et al., 2010*). We found that full removal of external calcium sources could further diminish the enhanced chondrogenic response obtained by thermo-mechanical cues when compared with TRPV4 inhibition alone. This behavior was consistent in gene expression profiles of the investigated markers and induced transcription returned toward control levels in the presence of EGTA. These findings indicate that calcium is a major signaling mechanism for thermo-mechanotransduction. Intracellular calcium oscillation is a key signaling mechanism in chondrocytes and can be triggered by different types of mechanical stimuli such as strain (*Lv et al., 2018*; *Han et al., 2012*) and fluid flow (*Degala et al., 2011*). Additionally, the dynamic characteristics of calcium signaling (e.g. amplitude and duration of $Ca^{2+}$ transients) were shown to be temperature dependent (*Han et al., 2012*). Our results suggest that the signaling cascade of thermo-mechanotransduction process is almost incomplete without external calcium sources. Previous studies also showed that extracellular $Ca^{2+}$ source is necessary for calcium signaling processes and increase in cytosolic $Ca^{2+}$ is not possible without extracellular $Ca^{2+}$, even when intracellular stores are full of calcium (*Pingguan-Murphy et al., 2006*; *Lv et al., 2018*).

When thermo-mechanical cues are applied, downregulation of Agc1 and COMP and upregulation of Twist1 were significantly different in the presence of EGTA compared to the condition in which TRPV4 antagonist was used (see *Figure 4*). This different response could arise from two plausible reasons. First, TRPV4 channels were not fully blocked by GSK 205 and the conveyed signal was strong enough to partially activate some of the channels. This is consistent with other studies (*Zhou et al., 2019*; *Lv et al., 2018*) where quantitative results indicated that 10 μm GSK205 significantly inhibited but did not fully abolish TRPV4 channel activity. A second possibility would be that, parallel thermo-mechanically responsive pathways besides TRPV4 were involved in $Ca^{2+}$ mediation in order to make this process more effective and robust. This is not surprising as accumulating evidence in the literature suggests that distinct calcium pathways, including stretch-activated channels (e.g. PIEZO1), voltage-gated calcium channels (e.g. T-type VGCC), purinergic receptors (e.g. P2Y or P2X), PLC-IP3-induced endoplasmic reticulum, and TRP family channels (e.g. TRPV3), are directly or indirectly influenced by an externally applied stimulus (*Pingguan-Murphy et al., 2006*; *Lv et al., 2018*; *Servin-Vences et al., 2017*; *Krupkova et al., 2017*). Determining other probable contributing mediators in thermo-mechanotransduction processes are thus important next steps and need further investigation.

In summary, different aspects involved in cartilage thermo-mechanobiology following joint loading were analyzed by developing a novel customized in vitro model. Our findings have demonstrated the superior effect of thermo-mechanical cues on chondrogenesis compared to either cue alone. Indeed, the applied synergetic stimulus provided a chondro-inductive signal by simulating the cartilage native environment under loading. This study also has shown that biomimetic temperature evolution as by-product of mechanical loading could itself induce biological responses of cells. Moreover, TRPV4 ion channels were identified as a key mediator of thermo-mechanotransduction process. Thermo-mechano-responsive nature of TRPV4 channels makes them a key target for future investigation on thermo-mechanobiology. We found calcium signaling as a contributing mechanism of action translating the effect of the externally applied cues to intracellular biochemical signals for cells. However, other pathways are also playing parallel roles, especially for Sox9 transcription, which therefore would necessitate further understanding of thermo-mechano-transduction processes overall.

# Materials and methods

## Key resources table

| Reagent type (species) or resource | Designation | Source or reference | Identifiers | Additional information |
|---|---|---|---|---|
| Cell line (Homo sapiens) | ECPs (human chondro-progenitor cells) | https://doi.org/10.3727/215517912X639324 https://doi.org/.10.3390/biom11020250 | | *Fetal chondro-progenitor cell source (FE002-Cart.) established after standardized processing of a fetal cartilage sample in Prof Lee Ann Applegate lab in 2009.* |
| Antibody | Anti-TRPV4 (Rabbit polyclonal) | Abcam | Cat#: ab191580 | IF (1:200) |
| Commercial assay or kit | Flou 4-AM (Calcium indicators) | Thermo Fisher Scientific | Cat#: F14201 | 5 μM |

*Continued on next page*

Continued

| Reagent type (species) or resource | Designation | Source or reference | Identifiers | Additional information |
|---|---|---|---|---|
| Chemical compound, drug | TRPV4 agonist; TRPV4 activator; GSK101 | Sigma-Aldrich | Cat#: G0798 | 10 nM |
| Chemical compound, drug | TRPV4 antagonist; TRPV4 inhibitor; GSK205 | AOBIOUS | Cat#: AOB1612-5 | 10 µM |
| Commercial assay or kit | NucleoSpin RNA XS | MACHEREY-NAGEL | Cat#: 740902.50 | |
| Commercial assay or kit | Live/dead (The Viability/Cytotoxicity Assay Kit) | Biotium | Cat#: 30002 | |
| Commercial assay or kit | PrestoBlue; PB (Cell Viability Reagent) | Thermo Fisher Scientific | Cat#: A13261 | |
| Software, algorithm | COMSOL Multiphysics | COMSOL Inc | COMSOL 5.4; RRID:SCR_014767 | |
| Software, algorithm | ImageJ | https://imagej.net/ | RRID:SCR_003070 | |
| Software, algorithm | MATLAB | MathWorks | RRID:SCR_001622 | |
| Other | EGTA | Sigma-Aldrich | Cat#: 03777 | |
| Other | RGD peptide (motif for cell attachment via integrin binding) | GL Biochem | | GGGRGDS-NH2 This peptide was synthesized by GL Biochem |
| Other | 4-Nitrophenyl chloroformate, NPC | Sigma-Aldrich | Cat#: 160210 | |
| Other | DAPI stain | Invitrogen | Cat#: D1306 | 2 µg/ml |
| Other | TRIzol Reagent | Thermo Fisher Scientific | Cat#:15596026 | |
| Other | Hoechst 33258 | Thermo Fisher Scientific | Cat#: H3569 | 0.2 µg/ml |
| Other | Loading machine | Instron | Cat#: ElectroPuls E3000 | |
| Other | PID microprocessor | Minco | Cat#: CT16A | |
| Other | Miniature RTD sensor | Minco | Cat#: S308 | |
| Other | Thermo-foil heater | Minco | Cat#: HM6975 | |
| Other | Gas mixer | ibidi | Cat#: 11922 | |

## In vitro culture and biological evaluations

ECPs were isolated from the proximal ulnar epiphysis of a 14-week gestation male donor (Ethics Committee Protocol # 62/07). It has been demonstrated that this progenitor cell source is very stable over many passages and gives reproducible behavior for chondrogenic differentiation (Darwiche et al., 2012). Cell growth medium consisted of DMEM containing L-glutamine, 4.5 g/l D-glucose and sodium pyruvate (Life Technologies) which was supplemented with 10% FBS (Sigma) and 1% L-glutamine and penicillin (Life Technologies). Cells were grown to 90% confluence, trypsinized and redistributed again in 2D culture up to passage 5 before seeding into scaffolds (~1.2 mio/scaffold). Stimulation medium was prepared with cell growth medium without FBS but having 10% ITS IV (Life Technologies) and 1% vitamin C. MTT staining Kit I (Roche) was used to control the cell distribution and Viability Assay Kit (Biotium) was utilized for live/dead evaluation according to the manufacturer's protocols. By using the MTT reagent, the purple formazan is produced due to the cells metabolic activity and therefore reveals cell distribution within the hydrogels. Viability assay was performed via a cell-permeable dye (calcine) for staining live cells and a cell-impermeable dye (ethidium homodimer) for staining dead cells having a damaged cell membrane. The Hoechst 33258 dye (Thermo Fisher Scientific) was employed for DNA quantification to evaluate the impact of hydrogel functionalization

on cell attachment. Briefly, the pure and RGD-modified hydrogels (with and without cells) were cut in small pieces and incubated overnight inside 500 ml papain digestion buffer at 65°C. Then, by dilution of 10 µl of digested solution in 140 µl of the dye (0.2 µg/ml), the emission signal of samples was measured at 460 nm by a Wallac microplate reader after excitation at 355 nm. The DNA content of the samples was finally determined by using a standard curve extracted from sequential dilutions of Calf Thymus DNA (Thermo Fisher Scientific) as calibrators. PrestoBlue viability kit (Thermo Fisher Scientific) was also used to monitor the cell proliferation inside the hydrogels during culture period (*Sonnaert et al., 2015*) based on manufacturer's protocol. Briefly, the PrestoBlue reagent was 10 times diluted in culture medium and samples were incubated inside. The fluorescent signal was then measured with Wallac 1420 VICTOR2 microplate reader for excitation/emission wavelengths of 544/590 nm, respectively.

## Design of modular bioreactor for thermo-mechanobiological study

The bioreactor chamber was custom designed and supplemented with different modules to reliably alter dynamic culture conditions. The mechanical design of the bioreactor chamber was optimized to ensure sterility and application of reproducible spatiotemporal stimuli on samples. The chamber, as shown in *Figure 2a*, contains 12 circularly arranged cylindrical wells for culture of cell-laden hydrogels. The set-up is compatible with the Instron uniaxial loading machine (Instron-E3000, Norwood, MA) to apply the mechanical loading (*Figure 2—figure supplement 5*) according to reported values for knee cartilage deformation during walking (*Liu et al., 2010*). The bioreactor was fixed to the Instron apparatus at its base by a support, and was stimulated from above by a set of pistons attached to the actuator. To control culture temperature, a thermo-foil heater (Minco-HM6975) was embedded below an aluminum conductive disc supporting culture wells. The temperature was measured by a miniature RTD sensor (Minco-S308) placed inside one of the wells. The evolution of culture temperature was feedback controlled by a PID microprocessor (Minco-CT16A, Minneapolis, MN) based on the error signal between the desired and monitored temperatures. The parameters of the PID controller were tuned experimentally based on the standard Ziegler-Nichols method. The heat controller was tuned to regulate the culture temperature defined by sequential ramps with different slopes according to the model prediction. This strategy was employed to simulate the temperature increase (occurring in knee joint with its irregular boundary conditions due to cartilage self-heating) in our small-size scaffolds (Ø:8, t: 2.2 mm) surrounded by the relatively large volume of culture medium (~2 ml) inside the bioreactor wells. To minimize the temperature gradient due to conduction within the sample from bottom to top, a medical graded thermoplastic part (PPSU) was incorporated within the stainless-steel wells. The geometry of PPSU part was carefully designed to make convection the dominant heat transfer mechanism to samples. A flexible silicon support was also employed to keep the position of samples on the center during dynamic loading. To maintain sterility, the wells were isolated from the outside environment with perforated solid caps covered with gas-permeable filters (OpSite Flexigrid, Smith & Nephew, Hull, UK). The $CO_2/O_2$ concentration and humidity inside the chamber were regulated by an external gas mixer (ibidi-Gas Incubation System, Martinsried, Germany) injecting humidified gas into the bioreactor. The humidity and gas concentration was directly measured inside the chamber with external calibrator probes to ensure correct functioning of the gas regulator, humidifier, and embedded sensors of the device.

## Dissipative hydrogel fabrication and mechanical characterization

Salt leaching method was employed for porous hydrogel fabrication by polymerization of HEMA-EGDMA precursor (4.8% molar ratio) inside molds containing sieved salt particles (150–250 µm) as described previously (*Nasrollahzadeh et al., 2019*; *Nasrollahzadeh and Pioletti, 2016*). Energy dissipation was determined by calculating the hysteresis area of stress-strain curve during loading and unloading of cycle 100. Equilibrium Young's modulus (Eeq) was measured by application of sequential stress relaxations and finding the slope of the corresponding relaxed stress in range of 10–20% strain values.

## RGD functionalization of hydrogels

The functionalization of pHEMA porous hydrogels with RGD peptide was performed following a published protocol with slight change (*Tugulu et al., 2007*; *Figure 2—figure supplement 2*). The

pendant hydroxyl groups of pHEMA hydrogels were firstly activated by 4-nitrophenyl chloroformate (NPC) and then RGD peptides were conjugated. For this purpose, lyophilized hydrogels were vacuum-swelled in the freshly prepared activation mixture of 1.41 g NPC in 15 ml acetone and 1 ml distilled triethylamine. The activation process was continued by agitation of samples in the solution for 1 hr at room temperature. The samples were then extensively rinsed with acetone and methanol several times to remove excess NPC. Afterward, the NPC-activated samples were transferred to the methanol solution containing 1 mM RGD peptides and 2 mM 4-dimethylamino-pyridine. To complete the functionalization process, the samples were incubated overnight at room temperature under gentle shaking in the reactor covered with aluminum foil. The RGD functionalized samples were then rinsed three times with pure methanol for 2 min. To ensure removal of any non-reacted NPC group, the samples were incubated in 0.5 M solution of ethanolamine in methanol for 20 min under gentle shaking. The samples were finally rinsed several times with methanol and bi-distilled water to wash non-specifically adsorbed peptides and entrapped NPC molecules inside the porous hydrogels.

**Table 1.** Material properties used in the heat transfer model of the hydrogel in the culture well.

| Material properties | Value | Unit |
|---|---|---|
| Equilibrium modulus ($E_{eq}$) | 500 | kPa |
| Poisson ratio ($\nu$) | 0.23 | – |
| Porosity ($\phi$) | 68 | % |
| Permeability (k) | $2 \cdot 1e^{-14}$ | $m^2$ |
| Dissipative power of hydrogel | 9000 | $w/m^3$ |
| Heat capacity of pHEMA ($C_{solid}$) | 1308 | J/(kg·kelvin) |
| Conductivity of pHEMA ($K_{solid}$) | 0.25 | W/(m·kelvin) |
| Heat capacity of water ($C_{fluid}$) | 4200 | J/(kg·kelvin) |
| Conductivity of water ($K_{fluid}$) | 0.6 | W/(m·kelvin) |
| Biot–Willis coefficient (α) | 1 | – |
| Relaxation modulus ($G_1$) | 200 | kPa |
| Relaxation modulus ($G_2$) | 58 | kPa |
| Relaxation modulus ($G_3$) | 215 | kPa |
| Relaxation time ($\tau_1$) | 0.42 | s |
| Relaxation time ($\tau_2$) | 5.82 | s |
| Relaxation time ($\tau_3$) | 1600 | s |

## Simulation of heat transfer inside bioreactor culture well

To ensure independent control over applied loading and temperature evolution in bioreactor culture wells, we developed a finite element (FE) heat transfer model using COMSOL Multiphysics software (Burlington, MA). The hydrogel dissipation was considered as a heat source generating 9000 W/m³ (determined experimentally based on area of hysteresis loop at 1 Hz) in each step of cyclic compression. The solid and the fluid phases of the porous scaffold were coupled based on theory of poroelasticity and a linear viscoelastic behavior for the solid part was assumed as described in our previous work (*Nasrollahzadeh et al., 2019*). Moreover, the heat transfer module was coupled to the resultant pressure/velocity field in porous media to consider convective heat transfer mechanism. In the model, the 2D axisymmetric cross section of cylindrical samples (2.2 mm thickness and 4 mm radius) and culture medium were divided into FEs and boundary conditions were applied as shown in *Figure 2—figure supplement 6* and *Figure 2—figure supplement 7*. The cyclic displacement (10% strain @ 1 Hz) was applied on the interface of the piston with sample-medium domains and stress, strain, velocity, pressure, and temperature fields were solved in a fully coupled manner. The model was examined for different mesh sizes to ensure that results are not changed with elements size. We assumed isotropic mechanical behavior for the poro-viscoleastic scaffold with material properties according to *Table 1*.

## Cartilage loading-induced self-heating in vitro

Human epiphyseal chondro-progenitor cells were expanded in 2D culture according to standard cell culture protocols (*Darwiche et al., 2012*) before seeding them into porous hydrogels. Hydrogels were disinfected by graded ethanol. Cells (~1.25 million per sample) were infused into them by an optimized compression released induced suction method (*Nasrollahzadeh et al., 2017*). The cell-seeded construct was prepared as a batch and then randomly distributed in different study groups. The effect of the different biophysical stimuli was then assessed by application of respective signals on cell-laden

**Table 2.** Primers data used for quantitative RT-PCR.

| Primer | Sequence | Concent. | Efficiency | Temp. |
|---|---|---|---|---|
| RPL13-F | TAAACAGGTACTGCTGGGCCG | 150 ng | 96% | 60 °C |
| RPL13-R | CTCGGGAAGGGTTGGTGTTC | | | |
| Agc-F | GGTACCAGTGCACAGAGGGGTT | 175 ng | 99% | 62 °C |
| Agc-R | TGCAGGTGATCTGAGGCTCCTC | | | |
| Twist-F | AGCAGGGCCGGAGACCTAGATGTCA | 250 ng | 95% | 60 °C |
| Twist-R | ACGGGCCTGTCTCGCTTTCTCT | | | |
| Comp-F | TGCTTCGGGAACTGCAGGAAAC | 250 ng | 101% | 60 °C |
| Comp-R | GCACGCGTCACACTCCATCACC | | | |
| SOX9-F | TGGAAACTTCAGTGGCGCGGA | 225 ng | 108% | 64 °C |
| SOX9-R | AGAGCAAAAGTGGGGGCGCTT | | | |
| COL1A1-F | CCTGCGTACAGAACGGCCTCA | 150 ng | 88% | 60 °C |
| COL1A1 -R | CGTCATCGCACAACACCTTGCC | | | |
| Col2a- F | GGAATTCGGTGTGGACATAGG | 175 ng | 96% | 60 °C |
| Col2a- R | ACTTGGGTCCTTTGGGTTTG | | | |
| TRPV4-F | TCCACCCTATATGAGTCCTCG | 250 ng | 99 | 60 |
| TRPV4-R | TAGGTGCCGTAGTCAAACAGT | | | |

hydrogels. The reported in vivo data in the literature for deformation of tibio-femoral cartilage during walking (*Liu et al., 2010*) and temperature rise during jogging (*Becher et al., 2008*) activities were used to apply respective stimulus in vitro. After the seeding step, all samples were pre-cultured for 4 days in cell growth medium inside standard incubators (32°C or 37°C and 5% $CO_2$). After this step, a cell stimulation medium was used and the thermal (32–39°C), mechanical (10% pre-strain, 10% amplitude at 1 Hz), or thermo-mechanical (their combination) stimulation were applied on samples starting from day 6. Three intermittent stimulations in alternate days were applied on the treated samples for 90 min while the control group was cultured in equivalent conditions but without receiving any stimulus (*Figure 3—figure supplement 2*).

## RNA isolation and real-time quantitative RT-PCR

Total RNA was extracted using the NucleoSpin RNA (Macherey-Nagel) after preparation steps. Briefly, samples were put in a 2 ml Eppendorf tube containing 300 μl Trizol. Hydrogels were disrupted with a polytron (Kinematica AG, Switzerland), while keeping them cold on dry ice. Then, 100 μl chloroform was added and samples were centrifuged for 5 min at 12,000 rpm at 4°C. The aqueous phase was transferred to 2 ml phase lock tubes and centrifuged for an additional 5 min at 12,000 rpm at 4°C. The aqueous phase was carefully transferred to 1.5 ml Eppendorf tubes and the extraction was completed by adding a RNA carrier and following the XS kit protocol. The RNA was quantified using the Nanodrop Lite Spectrophotometer (Thermo Scientific) and reverse transcription of 1 μg RNA was carried out using Taqman Reverse Transcription Reagents (Applied Biosystems, Bedford, MA). Fast SYBR Green Master Mix (Applied Biosystems) was used for PCR amplification in a final volume of 20 μl containing 10 ng of synthesized cDNA. The PCR amplification was performed for each sample by StepOnePlus Real-Time PCR device (Applied Biosystems) using specific human primers for different genes (*Table 2*). The cycling steps were defined as an initial 95°C for 2 min followed by 40 cycles of amplification. Gene expression data were analyzed using the comparative ΔΔCt (*Livak and Schmittgen, 2001*) method with RLP13a as the reference gene (*Studer et al., 2017*).

## Immunostaining

Characterization of TRPV4 protein on human chondro-progenitor cells was conducted by employing an anti-TRPV4 antibody (ab191580, Abcam) via standard immunostaining technique. Briefly, samples

were fixed by PAF (4%) and cells were permeabilized by triton (0.25%). After rinsing by PBST, samples were incubated for 1 hr in blocking solution (1% BSA) at room temperature and then overnight in TRPV4-specific antibody (5 µg/ml) at 4°C. Finally, the secondary antibody (goat-anti rabbit) conjugated with Alexa Fluor 488 was used to visualize TRPV4 channels by a confocal fluorescent microscope (LSM 700-Zeiss).

## Calcium imaging

Calcium imaging was performed by staining samples with 5 µM Flou 4-AM reagent (F14201, Thermofisher) according to manufacturer's protocol. Activation of TRPV4 channels was assessed by using 10 nM GSK101 in HBSS solution. The samples were imaged by spinning disc confocal microscopy (Visitron) up to 10 min and the effect of TRPV4 agonist was analyzed. To diminish total extracellular calcium and inhibit TRPV4-mediated calcium influx, free calcium ions were chelated with 2 mM EGTA and blocked the channel with 10 µM TRPV4 antagonist, GSK205, respectively.

## TRPV4 pathway and calcium signaling study

Gene expression of cell-laden hydrogels in different groups was tested by using medium containing vehicular control, TRPV4 antagonist, and calcium chelator. The cell-seeded samples were prepared as a batch and then randomly distributed in different study groups. The samples were stimulated according to our standard thermo-mechanobiological experiment protocol (*Figure 3—figure supplement 2*) while the medium was supplemented with 0.2% DMSO as control vehicle, 10 µM GSk205, or 2 mM EGTA. These chemicals were added to the culture wells 1 hr before start of stimulation and removed half an hour after stimulation to limit exposure of cells to pathway antagonists. Afterward, the samples were washed twice by fresh medium and stimulation medium was added without chemicals.

## Statistical analysis

All biological and mechanical experiments were analyzed with at least three replicates per test condition. The statistical significance between different study groups was determined by Student's t-test (Welch assumption). Statistical comparisons were performed with GraphPad Prism software (GraphPad). The bar charts show mean and SD values. The box and whiskers charts show mean, median, min, and max values. Calcium imaging evaluations were performed at different time points each covering more than 50 cells per test condition during TRPV4 activation, inhibition, and $Ca^{2+}$ chelation.

# Acknowledgements

We thank Sandra Jaccoud, Theofanis Stampoultzis, Dr Nathalie Burri for their technical support. We acknowledge the contribution of the Polymers Laboratory of EPFL and in particular Prof Harm Anton Klok for his advice on functionalization of pHEMA hydrogels. We appreciate the support of EPFL mechanics workshop and in particular Marc Jeanneret in design of the bioreactor chamber. The service given by CIME facility at EPFL to prepare SEM images is also acknowledged. This work was supported by the Swiss National Science Foundation (#310030_149969/1 and #CR23I3_159301).

# Additional information

### Funding

| Funder | Grant reference number | Author |
|---|---|---|
| Swiss National Science Foundation | #310030_149969/1 | Dominique P Pioletti |
| Swiss National Science Foundation | #CR23I3_159301 | Dominique P Pioletti |

The funders had no role in study design, data collection and interpretation, or the decision to submit the work for publication.

## Author contributions
Naser Nasrollahzadeh, Conceptualization, Data curation, Formal analysis, Investigation, Methodology, designed the customized model system and performed the in vitro tests and conceptualized the study and designed the experiments. NN developed the scaffolds functionalization method and executed mechanical characterization and validation tests. NN conducted calcium imaging and heat transfer numerical simulations and contributed to the analysis of the results. NN designed and prepared the figures and wrote the manuscript., Project administration, Resources, Software, Validation, Visualization, Writing – original draft, Writing – review and editing; Peyman Karami, Investigation, Methodology, Visualization, Writing – review and editing, contributed to the analysis of the results, designed and prepared the figures and reviewed the manuscript; Jian Wang, Formal analysis, Investigation, Methodology, developed the scaffolds functionalization method; Lida Bagheri, Investigation, Validation, Visualization, Writing – review and editing, executed mechanical characterization and validation tests, designed and prepared the figures and reviewed the manuscript; Yanheng Guo, Formal analysis, Investigation, Software, conducted calcium imaging and heat transfer numerical simulations; Philippe Abdel-Sayed, Conceptualization, Writing – review and editing, contributed to the analysis of the results and reviewed the manuscript; Lee Laurent-Applegate, Methodology, Resources, Writing – review and editing, reviewed the manuscript; Dominique P Pioletti, Conceptualization, Funding acquisition, Project administration, Resources, Supervision, Writing – review and editing, conceptualized the study and designed the experiments, contributed to the analysis of the results, reviewed the manuscript and supervised the project

## Author ORCIDs
Naser Nasrollahzadeh ⓘ http://orcid.org/0000-0001-9094-6251
Dominique P Pioletti ⓘ http://orcid.org/0000-0001-5535-5296

## Ethics
Human subjects: The work involves human subjects as the employed cell source for this study was established in 2009 after standardized processing of a fetal cartilage sample obtained from the proximal ulnar epiphysis of a 14-week gestation male donor (Ethics Committee Protocol # 62/07). The process was registered under the Swiss Federal transplantation program and its dedicated biobank, complying with the laws and regulations within both framework programs and stipulates that the donor is anonymous. The mother donor provided written informed consent that the tissue could be used for development of biobanks for clinical research purposes. The resulting biobank is the property of one of the authors (LAL) and cells have been provided for academic purposes and has been mutually agreed to publishing through a Material Transfer Agreement.

## Decision letter and Author response
Decision letter https://doi.org/10.7554/eLife.72068.sa1
Author response https://doi.org/10.7554/eLife.72068.sa2

---

## Additional files

### Supplementary files
• Transparent reporting form

### Data availability
All data generated or analysed during this study are included in the manuscript and supporting files. Source data files for gene expression results of figure 3 and 4 are provided.

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
