## [Editor Report]

This is the first demonstration that temperature evolution in cartilage following joint loading alters chondrogenesis through activation of TRPV4 channels and downstream transcriptional effects. This new biology is consistent with the recently established role of temperature in theregulation of bone cells.

---

## [Decision Letter]

**Decision letter after peer review:**

Thank you for submitting your article "Temperature Evolution Following Joint Loading Promotes Chondrogenesis by Synergistic Cues via Calcium Signaling" for consideration by *eLife*. Your article has been reviewed by 2 peer reviewers, and the evaluation has been overseen by a Reviewing Editor and Mone Zaidi as the Senior Editor. The following individual involved in review of your submission has agreed to reveal their identity: Farshid Guilak (Reviewer #2).

Essential revisions:

1) The authors need to discuss whether these reactions change the surface morphology of hydrogel functionalized by RGD peptides and further influence the cell morphology? They may use SEM images to address this question.

2) In addition to Col2 and Aggrecan, they may also check s-GAG/aggrecan and COMP.

3) To confirm changes of chondrogenesis, the authors need to examine the ratio of Col2 to Col1.

4) The authors need to determine the time dependent TRPV4 expression.

*Reviewer #1 (Recommendations for the authors):*

1. The authors functionalized the hydrogel with RGD-peptide to improve the cell adhesion and spreading on the scaffold, so hydroxylation and RGD grafting protocols are performed. Whether these reactions change the surface morphology and further influence the cell morphology ? If more details about the surface morphology, such as SEM images can be provided, it will be enlightening.

2. Although gene expression is one of the direct evidences for inducing chondroblast differentiation, more detection methods, such as GAG expression can be provided to make it more convincing.

3. As the authors hypothesize, calcium signaling is a major mechanism of action in transduction of thermo-mechanical cues sensed by TRPV4 channels. However, the available data on calcium signaling pathways do not support this conclusion strongly.

4. It's better to numbering the Figures in SI section according to the sequence in which they appeared in the manuscript.

*Reviewer #2 (Recommendations for the authors):*

This is a excellent study that required complex setup and analysis to address a topic in cartilage mechanobiology that has generally been neglected. The goal of this work was to study the effects of mechanical loading on temperature changes in the joint, and how that might affect cell response to loading.

I think the work is a major breakthrough. There are some specific questions about the model system that could use further explanation or discussion to improve the impact of the work. The following minor issues would benefit from further explanations and addition to the Discussion.

1. It is not clear exactly what cells were used, how they were sourced, and why primary adult chondrocytes were not used. If they were from epiphysis, is it known if they are articular chondrocytes and not growth plate? If the sex was known, please report this also.

2. The expression of TRPV4 can change dramatically with differentiation of stem/progenitor cells. Was TRPV4 expression measured over time in the scaffold? How did TRPV4 expression and function change with temperature?

3. If this phenomenon occurs in native cartilage, why was a complex hydrogel and model system used, instead of loading explants of cartilage? Would you expect the same results in native cartilage? The gel structure seems to provide a complex environment at the cellular level since cells are embedded with large pores rather than individual encapsulated. What is the average pore size?

4. Images of the cells show that they are not rounded but that they are attached and spread on the scaffold. How would this be expected to influence. What was the ratio of col2 to col1, as a measure of chondrogenesis? It is referenced as Figure 3 but this only shows relative expression to the control group for each gene.

5. TRPV4 is generally regarded as an osmotically sensitive channel in chondrocytes. The preculture for 12 days should produce sufficient proteoglycan to allow for mechano-osmotic transduction. Was any histology performed on specimens to show protein levels of s-GAG/aggrecan, collagens, COMP, etc.? Were any loading experiments performed immediately after seeding to test the role of newly formed ECM on mechanical or thermal sensitivity?

There are some typos in the manuscript – Figure 3 caption "nucleus", Figure 4 caption "characterization" and "intracellular" are misspelled.

[Editors' note: further revisions were suggested prior to acceptance, as described below.]

Thank you for resubmitting your work entitled "Temperature Evolution Following Joint Loading Promotes Chondrogenesis by Synergistic Cues via Calcium Signaling" for further consideration by *eLife*. Your revised article has been evaluated by me.

The manuscript has been improved considerably with a thoughtful response to review critique, but there is one remaining issue that needs to be addressed. Reviewer 1 has made the following comment. Could you please address this either by an explanation, or preferably, by any data you may have on chondrogenesis?

"Only RT-PCR test was performed and cartilage related genes expression was provided, it was far away from the 'chondrogenesis', since it is generally accepted that gene expression does not equate to protein synthesis, let alone the RT-PCR test is highly volatile."

---

## [Author Response]

Essential revisions:1) The authors need to discuss whether these reactions change the surface morphology of hydrogel functionalized by RGD peptides and further influence the cell morphology? They may use SEM images to address this question.

As suggested, we evaluated the scaffolds surface morphology after RGD functionalization and compared the result with non-functionalized samples by scanning electron microscopy (SEM). While we could not observe a noticeable change on the surface morphology of the polymeric scaffolds after functionalization process, the SEM images showed that cells could attach to multiple binding points on the pores of samples with RGD motifs.

2) In addition to Col2 and Aggrecan, they may also check s-GAG/aggrecan and COMP.

The COMP expression has been already reported in the first version of manuscript.

The effect of applied stimulation on s-GAG synthesis which is a downstream process after gene transcription was not examined in our work but mentioned as a follow-up study in the first version of the manuscript.

With this regard, our group has recently examined GAG synthesis following application of mechanical, thermal and thermomechanical stimuli and observed the highest s-GAG/DNA accumulation when samples received combined stimuli. These results indicated that application of dynamic thermal and mechanical stimuli can significantly improve chondrocyte ability to secrete matrix components.

3) To confirm changes of chondrogenesis, the authors need to examine the ratio of Col2 to Col1.

In the first version of the manuscript, we have reported the relative expression of Col2 and Col1 separately and the results have shown a significant upregulation of Col2 following biophysical stimulation and almost constant expression of Col1 for the studied groups. Here, we now report in Figure 3—figure supplement 1B , the calculate Col2/Col1 fold-changes for stimulated samples relative to control group.

The obtained relative fold-change indicates that applied stimuli enhance progenitor cell chondrogenic expression regardless of the absolute level of Col2 and Col1 genes. The superior effect of combined thermo-mechanical stimulation compared to thermal and mechanical loading is also noticeable.

However, prolonged culture of ECP cells with growth factor supplements (at least 21 days) is needed to reach mature chondrogenic differentiation state where the absolute expression of Col2 could exceed the absolute expression of Col1 as reported by other studies with this cell-line.

4) The authors need to determine the time dependent TRPV4 expression.

We have measured time-dependent gene expression of TRPV4 channels in the scaffolds and confirmed their functional expression on cells over our study period. When cultured in scaffolds without any stimulation, we observed downregulation of the TRPV4 genes at day 6 and 10 compared to day 2. However, by application of mechanical and/or thermal stimuli, a significant upregulation in expression of TRPV4 gene was detected. We could also successfully detect the protein level expression of TRPV4 channels in scaffolds at different time points by immunostaining.

Reviewer #1 (Recommendations for the authors):1. The authors functionalized the hydrogel with RGD-peptide to improve the cell adhesion and spreading on the scaffold, so hydroxylation and RGD grafting protocols are performed. Whether these reactions change the surface morphology and further influence the cell morphology ? If more details about the surface morphology, such as SEM images can be provided, it will be enlightening.

As suggested, we evaluated the scaffolds surface morphology after RGD functionalization and compared the result with non-functionalized samples by scanning electron microscopy (SEM). While we could not observe a noticeable change on the surface morphology of the polymeric scaffolds after functionalization process, the SEM images showed that cells could attach to multiple binding points on the pores of samples with RGD motifs.

2. Although gene expression is one of the direct evidences for inducing chondroblast differentiation, more detection methods, such as GAG expression can be provided to make it more convincing.

The effect of applied stimulation on GAG synthesis which is a downstream process after gene transcription was not examined in our work but discussed in the first version of the manuscript.

In a recent work, our group has examined the GAG synthesis following application of mechanical, thermal and thermomechanical stimuli and observed highest s-GAG/DNA accumulation when samples received combined stimuli. These results indicated that application of dynamic thermal and mechanical stimuli can significantly improve chondrocyte ability to secrete matrix components.

We add this for the Reviewer information only as the complete results of the study are part of follow-up research accomplished with our Ph.D students which we aim to publish soon.

3. As the authors hypothesize, calcium signaling is a major mechanism of action in transduction of thermo-mechanical cues sensed by TRPV4 channels. However, the available data on calcium signaling pathways do not support this conclusion strongly.

We found that full removal of external calcium sources could further diminish the enhanced chondrogenic response obtained by thermo-mechanical cues when compared with TRPV4 inhibition alone. Specifically, downregulation of Agc and COMP and upregulation of Twist1 were significantly different in the presence of EGTA compared to the condition in which TRPV4 antagonist was used (see Figure 4). This different response could arise from two plausible reasons. First, TRPV4 channels were not fully blocked by GSK 205 and the conveyed signal was strong enough to partially activate some of the channels. This is consistent with other studies where quantitative results indicated that 10 µm GSK205 significantly inhibited but did not fully abolish TRPV4 channel activity. A second possibility would be that, parallel thermo-mechanically responsive pathways besides TRPV4 were involved in ca^2+^ mediation in order to make this process more effective and robust. This is not surprising as accumulating evidence in the literature suggests that distinct calcium pathways, including stretch-activated channels (e.g. PIEZO1), voltage-gated calcium channels (e.g., T-type VGCC), purinergic receptors (e.g., P2Y or P2X), PLC-IP3 induced endoplasmic reticulum and TRP family channels (e.g., TRPV3), are directly or indirectly influenced by an externally applied stimulus. Determining other probable contributing mediators in thermo-mechanotransduction processes are thus important next steps and need further investigation to understand the overall mechanisms.

The above discussion was already in the first draft of the manuscript.

4. It's better to numbering the Figures in SI section according to the sequence in which they appeared in the manuscript.

Thank you for your comment. We rearranged the SI figures accordingly as the Reviewer suggested.

Reviewer #2 (Recommendations for the authors):This is a excellent study that required complex setup and analysis to address a topic in cartilage mechanobiology that has generally been neglected. The goal of this work was to study the effects of mechanical loading on temperature changes in the joint, and how that might affect cell response to loading.I think the work is a major breakthrough. There are some specific questions about the model system that could use further explanation or discussion to improve the impact of the work. The following minor issues would benefit from further explanations and addition to the Discussion.1. It is not clear exactly what cells were used, how they were sourced, and why primary adult chondrocytes were not used. If they were from epiphysis, is it known if they are articular chondrocytes and not growth plate? If the sex was known, please report this also.

Thank you for your comment. We briefly reported the employed cell-line in the respective sections of manuscript and referenced the corresponding publications regarding standardization and characterization of this cell source for further reading of interested people.

2. The expression of TRPV4 can change dramatically with differentiation of stem/progenitor cells. Was TRPV4 expression measured over time in the scaffold? How did TRPV4 expression and function change with temperature?

We have measured time-dependent gene expression of TRPV4 channels in the scaffolds and confirmed their functional expression on cells over our study period. When cultured in scaffolds without any stimulation, we observed downregulation of the TRPV4 genes at day 6 and 10 compared to day 2. However, by application of mechanical and/or thermal stimuli, a significant up-regulation in expression of TRPV4 gene was detected.

We could also successfully detect the protein level expression of TRPV4 channels in scaffolds in different time points by immunostaining. The functionality of the channels were also verified in 2D culture over 12 days (data not shown) culture by administration of 10 nM TRPV4 agonist (GSK101).

3. If this phenomenon occurs in native cartilage, why was a complex hydrogel and model system used, instead of loading explants of cartilage? Would you expect the same results in native cartilage? The gel structure seems to provide a complex environment at the cellular level since cells are embedded with large pores rather than individual encapsulated. What is the average pore size?

In addition to cartilage dissipation, other factors such as ambient temperature and heat transfer from surrounding tissues could contribute to the temperature evolution in the knee joint. Therefore, it is unlikely to have the same temperature evolution if we load the cartilage explants in a conventional compression bioreactor. Given the volume of the culture medium in each well, the size of the sample and the system boundary conditions, the accumulated lost energy from the dissipative test samples could negligibly change the temperature (<0.5°C) in an in vitro system. Our heat transfer simulation confirmed this observation when the well interface temperature was kept constant at 32°C (Figure 2—figure supplement 6). However, in an unrealistic adiabatic condition, our numerical simulation showed a temperature rise (Figure 2—figure supplement 7) due to hydrogel dissipative capacity (see supplemental info for detail). Therefore, the developed modular bioreactor is necessary for evaluating the role of temperature rise during mechanical loading in a laboratory design. Yet, the use of human articular cartilage explants instead of cell-seeded constructs in our customized bioreactor can better reproduce the in vivo thermomechanical environment. However, the difficulty in controlling healthy human articular cartilage with similar size and physiological state was the main reason for which we developed a reproducible model and used cell-laden hydrogels. Moreover, we did not use animal explants as the recorded temperature rise obtained for human volunteers during physical activity and such data for temperature rise inside intra-articular cavity of animals was not available. This temperature evolution rate was shown to be important in this manuscript.

The pore size distribution is mainly between 100-200 µm as shown in Figure 2—figure supplement 1.

4. Images of the cells show that they are not rounded but that they are attached and spread on the scaffold. How would this be expected to influence. What was the ratio of col2 to col1, as a measure of chondrogenesis? It is referenced as Figure 3 but this only shows relative expression to the control group for each gene.

As known, one mechanism by which cells may sense the externally applied stress is through their integrin interaction with anchored ligands within the extra cellular matrix. Despite controversial roles of ECM-mimetic elements in modulation of cell chondrogenic differentiation in free swelling hydrogels, a positive influence was observed for cell-laden hydrogels containing RGD in presence of dynamic loading . In particular, it is suggested that the moderate attachment of cells to hydrogel can make preculture time unnecessary for pericellular matrix formation before mechanical stimulation (48).

It has already been mentioned in the first version of the manuscript that cells could detach in non-modified scaffolds when cyclic compression was combined with temperature rise. This could lead to poor RNA extraction (low amount and quality) after multiple biophysical stimuli. We therefore employed ECM-mimetic functionalization of synthetic hydrogels to enhance cells adhesion and mechano-sensing during stimulation.

Despite providing a round morphology and chondro-inductive environment, encapsulation of cells in conventional hydrogels (alginate, agarose, gelatin, hyaluronic acid, etc) leads to cell-laden constructs with low stiffness dissimilar to cartilage properties. The ideal construct would be a mechanically capable scaffold that provides an environment for moderate cell adhesion and could maintain round morphology of cells. This might be realized by cell encapsulation into an adhesive soft hydrogel with endogenous binding sites for cells which is firmly integrated in a stiff and porous 3D support with load-bearing capacity . It is also suggested that covalent attachment between the soft and stiff components of the construct is important to properly transfer applied stimulus to cells. The limitation of our porous dissipative hydrogel for this method of construct preparation is its intentional low permeability (order of E^-14^ m^2^) to induce frictional drag dissipation. The permeability can be increased by enlarging the size of the pores and providing full interconnectivity within the porous structure. However, in this condition recapitulating cartilage dissipation cannot be achieved. The focus of current research was studying effect of loading induced temperature evolution on cell behavior by a reasonable in vitro model. Indeed, this system has its own limitation and the spread morphology of the cells in our dissipative porous hydrogels is one of them. Given the fact that we employed identically prepared cell-scaffolds constructs for all of our experimentation, we assumed that the isolated role of externally applied stimuli on cell chondrogenic response could be evaluated irrespective of cell morphology.

5. TRPV4 is generally regarded as an osmotically sensitive channel in chondrocytes. The preculture for 12 days should produce sufficient proteoglycan to allow for mechano-osmotic transduction. Was any histology performed on specimens to show protein levels of s-GAG/aggrecan, collagens, COMP, etc.? Were any loading experiments performed immediately after seeding to test the role of newly formed ECM on mechanical or thermal sensitivity?

The effect of applied stimulations on s-GAG synthesis which is a downstream process after genes transcription was not examined in our work but discussed in the first version of the manuscript.

In an unpublished recent work, our group has examined the GAG synthesis following application of mechanical, thermal and thermomechanical stimuli and observed highest s-GAG/DNA accumulation when samples received combined stimuli. These results indicated that application of biomimetic thermal and mechanical stimuli can significantly improve chondrocyte ability to secrete matrix components. We add this for the Reviewer information only as the complete results of the study are part of follow-up research accomplished with our Ph.D students which we aim to publish soon.

We did not apply the stimulation immediately after seeding to compare with current results. However, it is suggested that the moderate attachment of cells to hydrogel can make preculture time unnecessary for pericellular matrix formation before mechanical stimulation (48).

There are some typos in the manuscript – Figure 3 caption "nucleus", Figure 4 caption "characterization" and "intracellular" are misspelled.

Thank you for your comments. We corrected the misspelled words a double checked for all errors.

[Editors' note: further revisions were suggested prior to acceptance, as described below.]

The manuscript has been improved considerably with a thoughtful response to review critique, but there is one remaining issue that needs to be addressed. Reviewer 1 has made the following comment. Could you please address this either by an explanation, or preferably, by any data you may have on chondrogenesis?"Only RT-PCR test was performed and cartilage related genes expression was provided, it was far away from the 'chondrogenesis', since it is generally accepted that gene expression does not equate to protein synthesis, let alone the RT-PCR test is highly volatile."

The novelty of the current research is on the presentation of thermo-mechanobiology concept and development of an in vitro model system to study this coupled biophysical cue and associated mechanism of action. It is known that the reciprocal relationship between cells and ECM governs cartilage hemostasis as chondrocytes receive physical cues and regulate tissue turnover to maintain its functional characteristics. Chondrogenesis covers different stages of differentiation during chondrocyte maturation and a promoted gene expression is its molecular level reference. An altered gene expression is generally considered as the first indicator for cells responsiveness to conveyed cues which could ultimately lead to long term matrix deposition depending on other contributing factors. Our customized model system enabled us to show that chondrogenic cells immediately responded to biomimetic thermo-mechanical cues by altering their gene expression and that cells are able to sense the transient temperature rise. We evaluated cellular reactions after three rounds of stimulation to consider their probable desensitization and estimate a robust transcriptional response. However, given the transient nature of the chondrogenic markers to an applied stimulation, analysis of results at different time points could help to better understand the gene expression profile. Furthermore, it still remains to be determined to what extent the promoted chondrogenic gene expression could enhance downstream protein expression and functional properties. The workload of additional experiments and analysis to evaluate protein expression and functional properties is indeed out of scope of the current manuscript. However, as mentioned in the previous response to reviewer letter , our group has recently examined GAG synthesis following application of mechanical, thermal and thermomechanical stimuli and observed the highest s-GAG/DNA accumulation when samples received combined stimuli. These results indicate that application of dynamic thermo-mechanical stimulus can significantly improve chondrocyte ability to secrete matrix components following enhanced expression of chondrogenic genes.